# An Overview of Selected Rare B-Cell Lymphoproliferative Disorders: Imaging, Histopathologic, and Clinical Features

**DOI:** 10.3390/cancers13225853

**Published:** 2021-11-22

**Authors:** Ahmed Ebada Salem, Yehia H. Zaki, Gamal El-Hussieny, Khaled I. ElNoueam, Akram M. Shaaban, Bhasker Rao Koppula, Mark Bustoros, Mohamed Salama, Khaled M. Elsayes, Kathryn Morton, Matthew F. Covington

**Affiliations:** 1Department of Radiology and Imaging Sciences, Utah University School of Medicine, Salt Lake City, UT 84123, USA; Ahmed.Salem@utah.edu (A.E.S.); Akram.Shaaban@hsc.utah.edu (A.M.S.); Bhasker.Koppula@hsc.utah.edu (B.R.K.); Kathryn.Morton@hsc.utah.edu (K.M.); Matthew.Covington@hci.utah.edu (M.F.C.); 2Department of Radiodiagnosis and Intervention, Faculty of Medicine, Alexandria University, Alexandria 21566, Egypt; yehia.zaky@alexmed.edu.eg (Y.H.Z.); khaled.Elnoueam@alex.med.edu (K.I.E.); 3Department of Medical Oncology and Nuclear Medicine, Faculty of Medicine, Alexandria University, Alexandria 21566, Egypt; Jamal.attia@alexmed.edu.eg; 4Division of Hematology and Medical Oncology, Weil Cornell Medicine, Cornell University, New York, NY 10021, USA; mab4033@med.cornell.edu; 5Department of Pathology, Mayo Clinic, Rochester, MN 55901, USA; Salama.Mohamed@mayo.edu; 6Department of Diagnostic Radiology, The University of Texas MD Anderson Cancer Center, Houston, TX 77030, USA

**Keywords:** updates, WHO classification of hematologic malignancies, lymphoproliferative disorders, rare B-cell lymphomas, 18F-FDG PET/CT

## Abstract

**Simple Summary:**

The updated 4th edition WHO classification of lymphoid malignancies released in 2016 contains pivotal new terminology and information that is important for radiologists to understand. In spite of these updates, some lymphoproliferative disorders included in this update have been rarely discussed in the radiology literature. Many of these disorders have distinct clinical and imaging features, overlapping with more common disorders, thus causing a delay in diagnosis and management. Early diagnosis of many of these disorders is key as many of these are potentially treatable because early intervention may be lifesaving. The purpose of this manuscript is to provide guidance for radiologists regarding certain rare variants of B-cell lymphoproliferative disorders, in terms of clinical, histologic, and imaging findings, as well as to incorporate the new and updated terminology of some disorders included within the 4th edition of the WHO classification of lymphoid neoplasm.

**Abstract:**

Lymphoproliferative disorders (LPD) are conditions characterized by the uncontrolled proliferation of B or T-cell lines. They encompass a wide spectrum of abnormalities, which may be broadly classified as reactive processes or malignant diseases, such as lymphoma, based on their cellular clonality and clinical behavior. While some of these disorders are rare, they may be encountered sporadically in clinical practice, causing diagnostic dilemmas owing to overlap in their clinical and imaging features with more common disorders. The updated 4th edition WHO classification of lymphoid neoplasms was released in 2016 to incorporate the rapid clinical, pathological, molecular biology and cytogenetic advances of some of these disorders. Despite these updates, very little information is presented in the literature from the radiology perspective. The aim of this article is to familiarize radiologists and other physicians with certain rare variants of B-cell lymphoproliferative disorders with a focus on imaging features of these disorders, as well as to provide an overview of some important updates contained within the new WHO classification of lymphoid neoplasms.

## 1. Introduction

Lymphoproliferative disorders (LPD) are heterogenous conditions characterized by the uncontrolled proliferation of B or T-cell lines. They encompass a wide spectrum of abnormalities with varied clinical, pathologic, and imaging features. They are broadly classified as reactive or malignant diseases based on their cellular clonality and clinical behavior and include lymphomas and other hematologic malignancies [1]. Lymphomas are the most common adult hematological malignancies that arise from the lymphoid lineage cells or their precursors. Lymphomas can be broadly classified as Hodgkin’s lymphoma (HL) and non-Hodgkin’s lymphoma (NHL). NHL can be further subclassified as either B-cell lymphoma (BCL) or T-cell lymphoma (TCL) based on the cell of origin. Virtually any organ can be involved at any stage of the disease. However, at onset, nodal and splenic involvement is more common in HL, while extranodal disease is more frequent in NHL. The incidence of extranodal involvement in NHL is rising due to various factors including increases in human immunodeficiency virus (HIV) and indolent viral infections such as Epstein–Barr virus (EBV) [2].

Imaging plays a pivotal role in the management of lymphomas, most notably when using 18F-FDG PET/CT. The use of 18F-FDG PET/CT changes the staging of disease in nearly one-third of cases, frequently resulting in tumor upstaging by detecting occult sites not seen on conventional imaging with modalities such as CT or MRI. 18F-FDG PET/CT can clinically help to further subclassify lymphomas into aggressive or indolent subtypes based on standardized uptake values (SUVs) and the heterogeneity of disease involvement. Aggressive lymphomas often display higher SUVs (>10–13) at baseline, with disease manifesting with focal increased metabolic activity in nodal or extranodal sites [3]. 18F-FDG PET/CT can also provide information about the most amenable sites for biopsy to accurately diagnose and classify lymphomas. 18F-FDG PET/CT can also suggest whether two separate clonal tumors coexist in the same patient, manifesting with multiple lesions showing separate patterns of SUVs on the same imaging exam, thus revealing a need to biopsy two different sites to confirm separate clonal tumors by histological and molecular analysis [4]. During therapy, treatment-induced change in metabolic activity allows for the assessment of early tumor response, even before a change in tumor volume is appreciated. 18F-FDG PET/CT can also help predict which lymphomas are most likely to achieve early complete metabolic response, thus obviating the need for further consolidation therapy or high-intensity treatment protocols, decreasing unnecessary risk of treatment toxicity [4] 18F-FDG PET/CT can also suggest lymphoma transformation by detecting an increase in SUVs of lymphomatous masses despite adequate treatment received, although definitive confirmation of disease transformation requires confirmation by histopathologic and molecular analysis. After completion of therapy, some aggressive lymphomas, such as classic Hodgkin’s lymphoma (CHL) or primary mediastinal large B-cell lymphoma (PMLBCL), may show residual masses containing necrotic and/or fibrotic tissue with residual occult neoplastic cells. 18F-FDG PET/CT has been shown to be useful in identifying active residual sites of disease by the demonstration of persistent metabolic activity within one or more lesions [3,4].

## 2. Changes Contained in the Updated 2016 WHO Classification of B-Cell Lymphoproliferative Disorders (B-Cell LPD)

In 2016, the World Health Organization (WHO) updated the 2008 classification of tumors of hematopoietic and lymphoid tissue. This new update accounts for significant advances in knowledge of lymphoma behavior since 2008 and clarifies the pathogenesis and molecular biology of different types of lymphoproliferative disorders including lymphomas [5,6]. Table 1 summarizes important new and updated terminology from the 4th edition of the WHO classification of B-cell LPD. It is also important for radiologists and other physicians to be aware of rare lymphoproliferative disorders, some of which are discussed in the 4th edition WHO classification, that may be occasionally encountered in clinical practice (Table 2).

## 3. Uncommon B-Cell Lymphoproliferative Disorders

### 3.1. Primary Central Nervous System Lymphoma with Attention to Lymphomatosis Cerebri: An Extremely Rare Diffuse B-Cell Lymphoma Variant

Primary central nervous system lymphoma (PCNSL), including the extreme rare variant lymphomatosis cerebri (LC), is a rare subtype of NHL with exclusive involvement of the central nervous system (CNS), representing 1% of all intracranial malignancies [7]. LC implies the involvement of at least three cerebral lobes or three anatomical locations of the CNS, infiltrating the cerebral cortex and white matter tracts by lymphomatous cells, and some feel that LC would be more accurately named as a diffuse PCNSL variant [8]. Historically, PCNSL has been associated with immunodeficiency [9]. However, the incidence of PCNSL is increasing in immunocompetent patients, particularly among immunocompromised patients over 60 years of age [10]. There is an association of PCNSL with Epstein–Barr virus (EBV) infection that may account for at least a portion of PCNSL cases in immunocompetent patients [11]. In the previous 2008 WHO classification, it was hypothesized that the increasing incidence of PCNSL was falsely increased due to increased surveillance artifacts; however, in the current 4th edition of the WHO, an increasing incidence of PCNSL over the past two decades has been confirmed [5].

The majority of PCNSLs including LC are diffuse large B-cell lymphomas (DLBCLs), with the remaining cases representing low-grade BCL (8%), Burkitt (5%), or TCL (2–3%) [8,12]. Microscopically, LC is seen as a highly proliferative compact population of blastic B-lymphocytic cells with large pleomorphic nuclei and distinct nucleoli with predominant infiltration of the white matter and neuronal tracts without the formation of definite tumor or masses as seen in classic PCNSL. These lymphocytes showed the typical angiocentric infiltration pattern, and tumor cells invaded the neural parenchyma with a diffuse growth pattern forming perivascular cuffs [12]. Unlike in PCNSL, cerebrospinal fluid (CSF) analysis from lumbar puncture is usually non-specific as LC rarely involves the meninges. LC can be classified into primary and secondary types based on differences in CNS infiltration at initial presentation. Primary LC displays CNS infiltration at presentation, whereas secondary LC shows progressive infiltration over time but no infiltration at presentation [8,12].

Clinically, in both PCNSL and LC, the median age of presentation is 65 years, and in PCNSL most patients present clinically with focal neurological deficits. However, the most common presenting symptoms of LC are the impairment of cognitive functions with rapidly progressing dementia, behavioral abnormalities, and, less commonly, focal neurological deficits. The diagnosis of LC is usually delayed as clinical manifestations of LC are vague and overlap with imaging features of other disorders. The prognosis of LC is usually poor given the rapidly progressive disease process, and most LC patients do not survive beyond six months of initial presentation [12].

The classic imaging appearance of PCNSL is a focal solitary mass or multiple masses within the brain parenchyma. On CT, PCNSL often shows homogenously hyperdense lesions due to high tumor cellularity with homogenous enhancement and a predilection for the deep gray matter and deep nuclei, periventricular trigon, and corpus callosum that often abuts the sub ependymal/ventricular or meningeal surfaces [9]. Involvement of the meninges is most commonly seen in secondary lymphoma but can also be seen in PCNSL [13]. On MRI, PCNSL tumors tend to be hypointense to isointense on T1-weighted sequences (T1W) and isointense to hyperintense on T2-weighted sequences (T2W) with intense enhancement on post-contrast images (Figure 1).

Special sequences such as apparent diffusion coefficient (ADC) maps help differentiate PCNSL from high-grade tumors such as glioblastoma multiforme (GBM). PCNSL demonstrates marked diffusion restriction on diffusion-weighted images (DWI) with lower ADC values compared to GBM due to tumor restriction in lymphoma cells. The main role of 18F-FDG PET/CT in CNS lymphoma is to differentiate PCNSL vs. disseminated systemic lymphoma involving the CNS [14]. Additionally, 18F-FDG PET/CT can help in distinguishing PCNSL from mimicking disorders such as hypometabolic intracranial opportunistic infections including CNS toxoplasmosis [15,16,17,18]. Imaging findings of LC include diffuse, non-mass-like lesions within both hemispheres involving the white matter, corticospinal tracts (CST), and deep gray matter. These lesions show patchy contrast enhancement and diffuse abnormal T2W hyperintensity. Unlike PCNSL, LC shows variable restriction on DWI [19]. Regions commonly affected in LC include the subcortical, deep, and periventricular white matter, CST, U-fibers, corpus callosum, and gray matter, with less frequent involvement of the spinal cord compared to PCNSL [12]. Contiguous spread of lesions from the cerebral white matter to the brain stem along the corticospinal tract (CST) is a notable feature of LC, which can involve both the brain and brain stem (Figure 2 and Figure 3). Unlike PCNSL, the characteristic neuroradiological findings of LC with 18F-FDG PET/CT have not yet been established; nevertheless, they are usually seen as hypermetabolic mass like lesions [8,20]. MRI spectroscopy (MRS) may help differentiate LC from other neurodegenerative or inflammatory conditions. On MRS, evidence of high cell membrane turnover (high choline peak), neuronal damage (decreased NAA levels), high lactate levels, and elevated lipids is typically demonstrated. Although many of these MRS features resemble high-grade gliomas and metastases, elevated lipids have been shown to be potentially useful to differentiate between these entities and LC [21]. The diagnosis of lymphomatosis cerebri should be considered in patients presenting with a rapidly progressive decline in cognitive function, dementia, or behavioral abnormalities. Imaging findings including diffuse bilateral hemispheric cortical spinal tract involvement and pathology may show predominant infiltration of the neuronal tracts without the formation of definite masses. It is crucial that diagnosis of LC is only made after the exclusion of other common etiologies as the clinical and imaging findings of LC can be misattributed to other diffuse infiltrative brain tumors, leukoencephalopathy, vasculopathy, degenerative disease, ischemic processes, infectious processes, and toxic demyelinating diseases in addition to dementias and other psychiatric disorders such as depression [8,12].

### 3.2. Autoimmune Lymphoproliferative Syndrome

Autoimmune lymphoproliferative syndrome (ALPS) is a lymphoproliferative disorder that may be interpreted as malignant lymphoma. ALPS results from inherited mutations in the apoptosis signaling pathways [22,23]. ALPS was first recognized in the early 1990s. Due to the overlap of clinical, pathologic, and imaging features with other malignant lymphoproliferative disorders, in addition to histopathology, a clinical diagnostic criterion for ALPS has been developed to help establish the diagnosis [23,24]. The diagnostic criterion for ALPS includes both required and accessory criteria. Required criteria include chronic nonmalignant, noninfectious lymphadenopathy and/or splenomegaly (>6 months) and elevated CD3 + TCRab in the setting of normal or elevated lymphocyte counts with a lack of CD4 or CD8 expression, which are referred to as double-negative T cells (DNTs). Accessory criteria include somatic or germ-line pathogenic mutation (FAS, FASLG), elevated biomarkers (e.g., vitamin B-12), elevated IgG levels (polyclonal hypergammaglobulinemia), and family history of a nonmalignant/noninfectious lymphoproliferation with or without autoimmunity [23,24]. A probable ALPS diagnosis can be entertained by the presence of the required criteria and any one of the secondary accessory criteria. Patients with probable ALPS undergo genetic or apoptosis assay to confirm the diagnosis [24].

Clinically, ALPS presents in childhood and results in massive hepatosplenomegaly, chronic lymphadenopathy, thymic enlargement, and, in most patients, autoimmune phenomena such as hemolytic anemia [25].

Imaging is not necessary to establish a diagnosis of ALPS but might aid in its diagnostic evaluation. Stability in the size of lymphadenopathy and hepatosplenomegaly over many years may suggest ALPS from other malignant lymphoproliferative diseases [26] (Figure 4).

Once the diagnosis of ALPS is established, baseline and periodic follow-up CT scans should therefore be obtained to document the stability of lymphadenopathy and hepatosplenomegaly. 18F-FDG PET/CT may also be helpful because mild to moderate FDG uptake is typical of ALPS compared to the high FDG uptake characteristic of malignant lymphomas. By evaluating for areas of high FDG uptake, 18F-FDG PET/CT may help in the early detection of lymphoma transformation and demonstrate the best site for biopsy, when indicated [23]. Awareness of ALPS is pivotal because ALPS often requires long-term immunosuppressive therapies as opposed to chemotherapy that is standard for malignant lymphoproliferative disorders.

### 3.3. Primary Mediastinal Large B-Cell Lymphoma

Primary mediastinal large B-cell lymphoma (PMLBCL) is an uncommon B-cell lymphoma, previously classified as a subtype of DLBCL but now considered an aggressive neoplasm that originates from thymic B cells, constituting less than 3% of all NHL cases [27]. Classic HL (CHL) is always part of the clinical and imaging differential diagnosis of PMLBCL given the histologic overlap and many shared clinical features.

The median age of presentation with PMLBCL is slightly older than CHL, commonly affecting middle-aged women in their third or fourth decade. PMLBCL has a more favorable survival than other subtypes of aggressive DLBCLs, and poorer prognosis compared to CHL. The 5-year survival rate seen in cases of PMLBCL is 85% [28]. Patients usually presents with local mediastinal compressive symptoms such as cough, chest pain, dyspnea, and vascular compressive symptoms such as superior vena cava syndrome. Factors that predict poor performance include male sex, old age, extension into the adjacent surrounding thoracic viscera such as pleural and pericardium, the presence of extra mediastinal extranodal disease (kidneys, adrenal glands, liver) at diagnosis, and/or early inadequate response to induction therapy [27,28].

The genetic profile of PMLBCL is more similar to CHL than other BCLs [3]. It is crucial for pathologists to be able to differentiate PMLBCL histologically. Microscopically, CHL histologic features usually show abundant infiltration with large, Hodgkin, and Reed–Sternberg cells (HRS) in a polymorphous background composed of small lymphocytes, macrophages, eosinophils, and plasma cells. HRS cells typically show expression of CD30 and CD15 and lack expression of CD20 and CD45. In PMLBCL, samples usually show diffuse proliferation of medium to large B cells associated with sclerosis and a degree of compartmentalization in more than 50% of cases. Lymphocytes usually express CD45 and CD3. CD15 is often absent and more heterogenous compared to HL. CD10 is often negative. Another differentiating feature is MAL protein positivity, which is a glycolipid-enriched membrane involved in the tumorigenesis of PMLBCL [27]. Cases that cannot be differentiated from CHL after an extensive immunohistochemical workup should be categorized in the newly designated category: B-cell lymphoma, unclassifiable, with features intermediate between DLBCL and CHL (previously designated as “Gray-zone lymphomas”) [28].

On imaging, patients typically present with a large ill-defined anterior mediastinal mass that often measures more than 10 cm in diameter with local infiltration of the underlying mediastinal structures [29,30,31]. On CT, PMLBCL displays mixed attenuation due to the presence of necrosis and hemorrhage with heterogeneous post-contrast enhancement. On MRI, PMLBCL masses may show low signal on T1WI and variable signal on T2WI. After receiving chemotherapy, tumors tend to show decreased size with low signal on both T1WI and T2WI due to a fibrotic response.

The differential diagnosis of PMLBCL includes other anterior mediastinal masses such as CHL, germ cell tumors, and large thymic tumors, which can display necrosis and calcifications on CT [32]. Differentiation between PMBCL and CHL is not possible based on imaging findings alone. More aggressive clinical behavior such as superior vena cava syndrome may help distinguish PMBCL from CHL [33] (Figure 5).

The use of 18F-FDG PET/CT is essential in the evaluation of patients with PMLBCL to reveal sites of disease not visible on CT and to provide more accurate staging and radiation field planning (Figure 6). 18F-FDG PET/CT may also be beneficial for restaging after chemotherapy and/or radiotherapy, or when relapse is suspected [34]. Negative 18F-FDG PET/CT after two or four cycles of chemotherapy has a negative predictive value and may predict excellent outcome in patients, achieving complete response without relapse. Patients who have residual activity equal to or higher than liver activity after immunochemotherapy treatment are more likely to relapse. In such instances, the addition of radiotherapy to the treatment regimen should be considered to avoid relapse in those high-risk patients. Relapse usually occurs within 1 year and is more likely to be widespread, involving distant extranodal sites such as the CNS, liver, kidneys, adrenal glands, GI tract, ovaries, and pancreas. Late relapses are very uncommon [35]. 18F-FDG PET/CT can also efficiently assess post-treatment response, differentiating between necrotic or fibrotic tissue and residual masses containing viable tumor [33,36]. There are various potential challenges to 18F-FDG PET/CT post-treatment implementation, including false-positive results secondary to thymic rebound hyperplasia, specifically seen in the young population. This can be limited by increasing the interval between treatment and imaging. Additionally, MRI can be helpful in those cases, and high signal on T1 in phase imaging with loss of signal on the out of phase sequences is consistent with thymic rebound hyperplasia (Figure 7 and Figure 8) [34,37].

### 3.4. Lymphomatoid Granulomatosis

Lymphomatoid granulomatosis (LG), also known as angiocentric lymphoma, is a rare lymphoproliferative disorder. It results from the uncontrolled proliferation of EBV-infected B cells with prominent T-cell reactivation. Liebow et al. [38] first described LG in 1972. LG can be viewed as a clinicopathologic spectrum that ranges from a benign lymphocytic inflammatory vasculitis to a more aggressive lymphoma disorder. Histopathological examination of LG is characterized by polymorphic lymphoid infiltration composed mainly of lymphocytes (primarily reactive T cells, with varying numbers of atypical B cells), plasma cells, and histiocytes [39]. LG can be classified in a three-scale grading system that has important treatment implications based on the number of EBV-infected B cells and cytological atypia [39]. Grade I is consistent with benign lymphocytic vasculitis and granulomatosis, grade II follows an aggressive course, and grade III should be considered DLBCL. The diagnosis of LG is challenging because the disease is extremely rare, and the histopathological findings can be quite complex [39].

LG is more common among immunocompromised patients, affecting males more than females, and is most common in the fifth or sixth decade of life [40]. Although it is considered a lymphoproliferative disorder, the lymph nodes, spleen, and bone marrow are usually spared until late in the disease. In almost all patients, the lungs are involved, and other sites of disease include the skin (seen in one-third of cases), CNS (also seen in one-third of cases) (Figure 9 and Figure 10), and less commonly renal and liver involvement is seen. The involvement of other organs has also been reported, such as the upper respiratory and gastrointestinal tracts [39]. Symptoms associated with LG are non-specific and include fever, weight loss, night sweats, cough, and chest pain. Symptoms can imitate various clinical disorders such as infectious processes like fungal and tuberculous infections [39].

Imaging findings of LG include bilateral diffuse reticulonodular opacities and mass-like lesions involving mainly the middle and lower lung zones with predominantly central involvement and relative sparing of the periphery. On CT, the reticulonodular opacities represent multiple small nodules that have irregular margins. Nodules typically show an interseptal and peribronchovascular distribution. Nodules tend to coalesce over time, forming larger masses that may manifest as coarse irregular opacities surrounded by ground-glass opacities.

The differential diagnosis for these imaging findings is broad and includes pulmonary sarcoidosis, lymphocytic interstitial pneumonitis (LIP), atypical infections, and hypersensitivity pneumonitis. LG is usually not associated with mediastinal nodal enlargement or air trapping and the clinical course tends to be rapidly progressive and non-responsive to antimicrobial treatment. Such findings may be beneficial in excluding sarcoidosis and LIP as well as atypical infections [41,42]. Synchronous regression and progression of the nodules at chest CT may be observed and should be taken into consideration in the evaluation of response to therapy [41].

With 18F-FDG PET/CT, LG may show central FDG uptake within pulmonary nodules and masses. Peripheral uptake in the walls of cavitary pulmonary lesions (marginally or eccentrically) has also been reported. The appearance of coarse linear opacities may represent a healing phase of nodules and masses [43]. LG should be considered as a diagnostic possibility for any patient presenting with such clinical and imaging features and may be subsequently confirmed by histopathologic evaluation. LG is a potentially treatable disease, and early diagnosis, with aggressive intervention, may be lifesaving. First-line treatment includes corticosteroids, either alone or in combination with cyclophosphamide [42,44] (Figure 9 and Figure 10). The diagnosis of lymphomatoid granulomatosis should be suspected in immunosuppressed patients presenting with lower lung predominant peribronchovascular or subpleural masses and nodules with additional cutaneous, CNS, and renal involvement. Disorders such as sarcoidosis, granulomatosis with polyangiitis, and atypical infections must first be excluded.

### 3.5. Lymphoplasmacytic Lymphoma, Waldenstrom Macroglobulinemia, and Bing–Neel Syndrome

Lymphoplasmacytic lymphoma (LPL) is a rare indolent BCL characterized by overlapping features between B-cell lymphoproliferative disorders and multiple myeloma (MM). LPL is characterized by IgM paraproteinemia and rarely IgG or IgA production. WM is usually reserved for patients presenting with LPL in addition to bone marrow infiltration. It remains controversial whether LPL and WM are different manifestations of a single disease or two unique entities [45]. WM is considered a rare malignancy with an incidence of 3–4 cases per million people in the United States [46]. The median age at diagnosis is 70 years and the disease shows a higher prevalence among white men. Most cases are sporadic; familial predisposition is seen in only 20% of patients [46]. Historically, monoclonal gammopathy of unknown significance (MGUS) was believed to be a premyelomatous precursor, developing before frank MM onset [47]. However, now MGUS is classified according to secreting immunoglobulins to non-IgM and IgM cases. Non-IgM MGUS represents the majority of cases and may progress to multiple myeloma; IgM cases, however, are the ones that are now considered a separate rare variant of NHL under the updated 4th edition of the WHO classification of hematological malignancies and are more closely related to LPL than MM [5].

Histologically, WM shows mixed small B lymphocytes and plasma cells known as lymphoplasmacytic cells [47]. Similar to MM, WM shows rouleaux phenomena on blood smear and serum and urine electrophoresis show elevated IgM levels. Definitive diagnosis requires bone marrow biopsy that shows more than 10% mixed lymphoplasmacytic cells [48]. Molecular analysis of many WM patients identified MYD88 ^L265P^ mutation as a driver event in the pathogenesis of WM that occurs in 95% of patients. This specific biomarker is currently used to distinguish this malignancy from other overlapping BCLs, such as marginal zone lymphoma, as these disorders all share elevated IgM levels [49]. LPL has a 10% risk of transformation into more aggressive lymphoma subtypes, such as DLBCL [50].

WM is characterized by a wide spectrum of clinical presentations varying from bone marrow and organ infiltration by malignant lymphoplasmacytic cells to elevated IgM paraprotein, causing anemia, hyperviscosity, extramedullary disease, and peripheral neuropathy. A substantial number of patients, however, are asymptomatic at the time of diagnosis. In the absence of features of other lymphoproliferative disorders, LPL is a diagnosis of exclusion that may be reached after the elimination of other small B-cell IgM-secreting lymphomas and MGUS or myeloma [51].

Multimodality imaging, including CT, MRI, and/or 18F-FDG PET/CT, is essential for initial diagnosis, especially in extramedullary disease presentation. Imaging can be helpful in determining optimal biopsy sites, as well as the assessment of response to therapy and monitoring disease progression or transformation. Unlike MM, WM is not associated with definite osseous lytic lesions, a feature that can distinguish WM from MM. WM usually manifests as osteopenia, marrow space widening, and endosteal erosions [52,53] (Figure 11).

WM patients can also present with splenomegaly with or without focal lesions and lymphadenopathy typically within the axillary, retroperitoneal, and/or inguinal nodal stations (Figure 12). Cranial nerve deficits can be a manifestation of Bing–Neel syndrome, a rare CNS manifestation seen in WM disease [45,48,52,54]. In Bing–Neel syndrome, neurological symptoms can be directly related to infiltration by lymphoplasmacytic cells or indirectly result from the deposition of circulating IgM immunoglobulin on the brain parenchyma, spinal cord, or cranial or peripheral nerves (Figure 13). Additionally, 18F-FDG PET/CT plays an important role in monitoring therapeutic response after treatment initiation, especially with rituximab. WM patients tend to show elevated IgM levels, known as an IgM flare, which may mimic progressive disease. Imaging plays a pivotal role to differentiate between an IgM flare and true progression, thus obviating the need for repeated marrow biopsy [52]. Imaging may also help in excluding the development of myelodysplastic syndromes, acute leukemia, or DLBCL transformation [52].

## 4. Conclusions

In 2016, the WHO released an update to its lymphoid neoplasm classification including various lymphoproliferative disorders, and this update is intended to account for the rapid changes in our understanding of lymphoma behavior, and to incorporate the increased high-end technology advances within this field, to study molecular drivers and cytogenetics of these disorders. Despite these updates, many of these disorders are only addressed in a few case reports within the radiology literature. Radiologists must stay abreast of these updates as they play a major role in the management of these disorders as they are often the first to suggest the diagnosis, help with disease staging, assess response to therapy, and identify disease relapse.

## Figures and Tables

**Figure 1 cancers-13-05853-f001:**
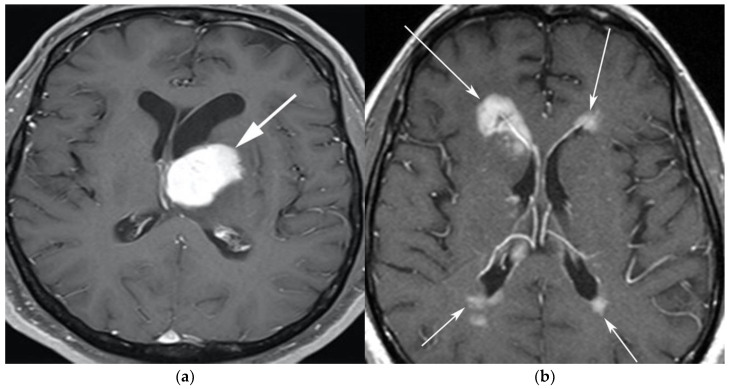
A 65-year-old man presenting with headache. (**a**) Axial T1 post-contrast fat-suppressed image shows homogenous enhancing mass within the left thalamus (white arrow) with mass effect on the 3rd ventricle and left putamen laterally. (**b**) Axial T1 post-contrast fat-suppressed images in a different patient show infiltrative periventricular soft tissue masses (white arrows). Both cases were pathologically proven primary central nervous system lymphoma-diffuse large B-cell histologic subtype.

**Figure 2 cancers-13-05853-f002:**
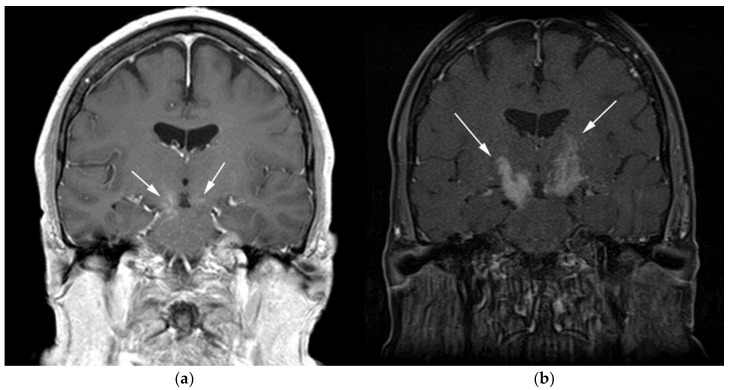
A 64-year-old-woman presenting with rapid onset of memory deterioration and altered sensorium. (**a**) Coronal MRI T1 post-contrast images show subtle enhancement along bilateral deep nuclei. At this time, no diagnosis was determined. After 3 weeks, patient presented to emergency department with worsening symptoms. A follow-up MRI was performed. (**b**) Coronal MRI post-contrast fat-saturated images revealed progressive increase in extent of diffuse infiltrative enhancing masses (white arrows).

**Figure 3 cancers-13-05853-f003:**
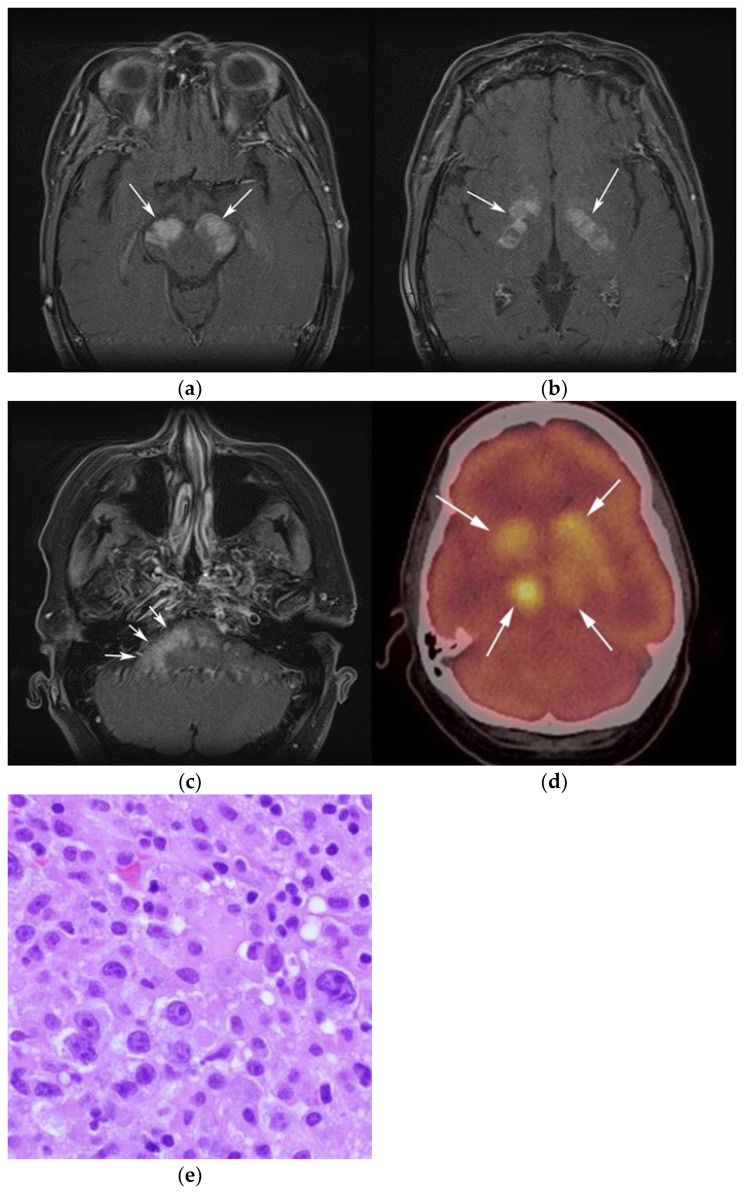
Same MRI series. (**a**) Axial post-contrast MRI image shows lesions extending caudally to involve the bilateral superior cerebral peduncles and anterior aspect of the mid-brain. (**b**) Axial MRI T1 post-contrast fat-saturated images show involvement of the posterior limb of the internal capsule and the thalami bilaterally (white arrows). (**c**) Extension along the lateral aspect of the pons, dentate nucleus, and the middle cerebellar peduncles (white arrows). (**d**) Axial FDG fused PET/CT image shows increased activity in the aforementioned lesions. Additional hypermetabolic lesions are seen along the course of the corticospinal tract (white arrows). Imaging findings and pattern of involvement are consistent with lymphomatosis cerebri. (**e**) H&E section shows a tumor composed of large and pleomorphic cells with intermingled small lymphocytes. Primacy CNS lymphoma typically demonstrates an angio-centric predilection. (Original magnification 400×, H&E stain). Other differential diagnoses include a vasculitic process, toxic or metabolic encephalopathy, paraneoplastic syndrome, or acute disseminated encephalomyelitis. Biopsy showed a large non-cohesive B-cell lymphocyte population consistent with LC.

**Figure 4 cancers-13-05853-f004:**
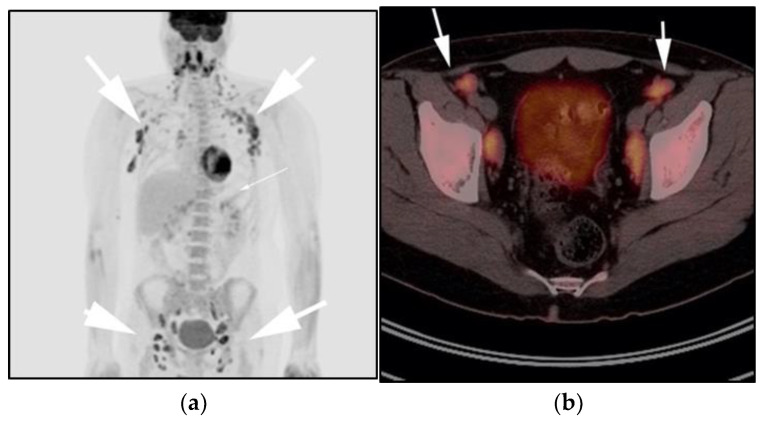
A 30-year-old male with known history of autoimmune lymphoproliferative disorder since childhood. (**a**) Maximal intensity projection (MIP) FDG PET/CT image shows generalized hypermetabolic lymphadenopathy throughout the head, neck, chest, abdomen, and pelvis (white arrows). Note that splenic activity is not visualized (thin white arrow) because the spleen was surgically removed during childhood due to splenomegaly. (**b**) Axial FDG PET image shows hypermetabolic adenopathy within pelvis mainly about the bilateral pelvic side wall and external iliac nodes (white arrows). Lymph nodes often show little or only modest uptake, a clue, in addition to stability of nodal size and over multiple time points. Biopsy is the only method for definitive diagnosis. (**c**) H&E section of a lymph node shows highly proliferative paracortical expansion composed of small proliferative lymphocytes and immunoblasts with preserved lymphoid follicular architecture. There is sinus histiocytosis. (Original magnification 400×, H&E stain).

**Figure 5 cancers-13-05853-f005:**
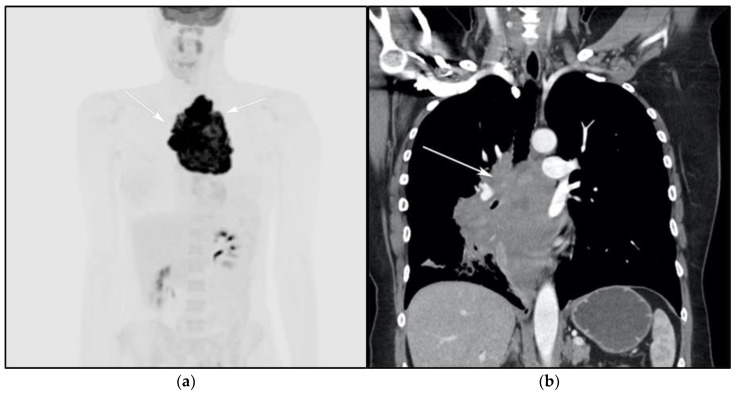
A 24-year-old female presenting with cough and dyspnea with plethora of the face. (**a**) Maximal intensity projection (MIP) FDG PET/CT shows hypermetabolic anterior mediastinal mass (white arrows) with no other sites of disease involvement in the body. (**b**) Coronal contrast-enhanced CT image shows significant vascular and airway compromise by an ill-defined anterior mediastinal mass (white arrow). Biopsy-proven primary mediastinal B-cell lymphoma. Main differential diagnosis would be Hodgkin’s lymphoma. However, unlike in Hodgkin’s lymphoma, primary mediastinal B-cell lymphoma often shows frequent invasion of the mediastinal vessels, frequently resulting in superior vena cava syndrome.

**Figure 6 cancers-13-05853-f006:**
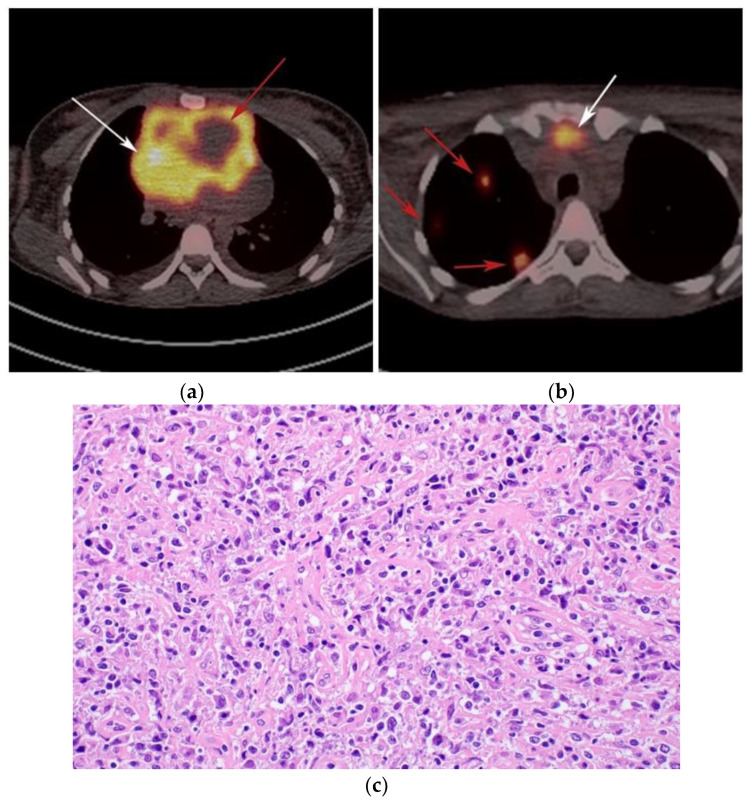
Imaging in a different patient with known diagnosis of primary mediastinal B-cell lymphoma. (**a**) Axial FDG PET/CT images show the hypermetabolic anterior mediastinal mass (white arrows), and there is also central areas of absent metabolic activity within the mass (red arrow), correlating with areas of fibrosis, a finding that is essential in prebiopsy planning to avoid false negative results. (**b**) Axial FDG PET/CT images show hypermetabolic right upper paratracheal nodes (white arrows) and several additional hypermetabolic pulmonary and pleural-based nodules within right upper lung (red arrows). (**c**) H&E section shows that the tumor is composed of large, atypical cells with reniform or multi-lobulated nuclei with abundant clear cytoplasm. Note the lymphoma cells are compartmentalized by the prominent sclerotic bands of fibrosis. (Original magnification 400x, H&E stain).

**Figure 7 cancers-13-05853-f007:**
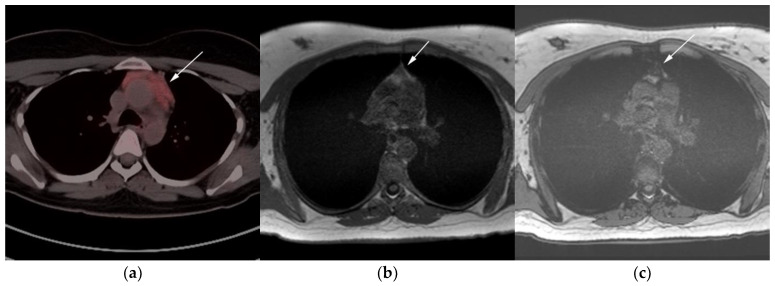
Imaging in a 20-year-old female with history of classic Hodgkin’s lymphoma (CHL) who had 4 cycles of chemotherapy. (**a**) Axial 18F-FDG PET/CT showing residual wedge-shaped/triangular activity seen within the anterior mediastinum (arrow). There was concern for residual disease vs. thymic rebound hyperplasia. (**b**) MRI axial T1 in phase and (**c**) MRI axial T1 out of phase show intermedial signal on T1 in phase (arrow) with dropped signal on the out of phase sequence (arrow), consistent with fat content due to thymic rebound hyperplasia.

**Figure 8 cancers-13-05853-f008:**
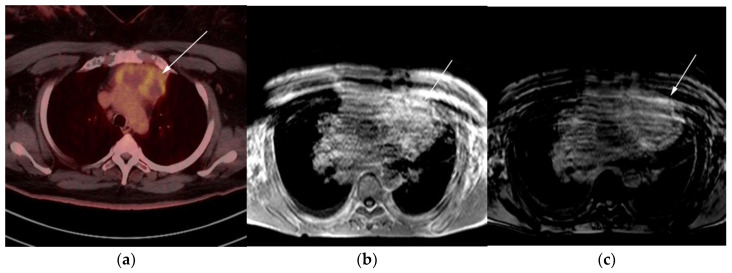
Imaging in a 60-year-old female presenting with ill-defined anterior mediastinal mass. (**a**) Axial 18F-FDG PET/CT showing ill-defined hypermetabolic mass within the anterior mediastinum (arrow). This was biopsy-proven thymic carcinoma. (**b**) MRI T1 in phase sequence, (**c**) MRI T1 out of phase—note there is no dropped signal on the T1 out of phase (arrows) due to lack of fat. This is consistent with residual disease.

**Figure 9 cancers-13-05853-f009:**
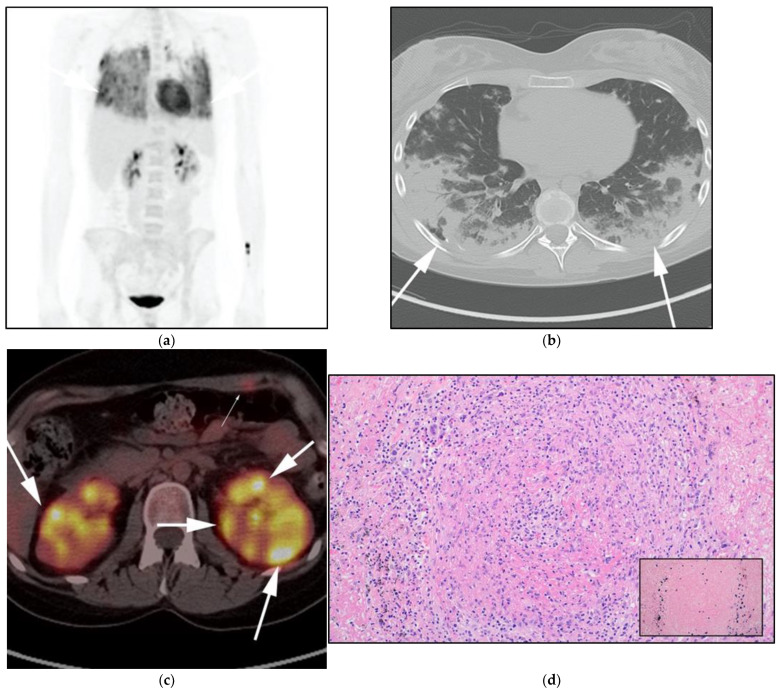
A 20-year-old-female with known diagnosis of lymphomatoid granulomatosis (LG) with history of rheumatoid arthritis. FDG PET/CT was ordered for staging. (**a**) Maximal intensity projection (MIP) FDG PET/CT image shows patchy hypermetabolic opacities involving the lower and mid lung zones (white arrows). (**b**) Axial CT image of the chest shows corresponding confluent consolidative patchy consolidation containing air bronchograms as well as numerous consolidative nodules (white arrows) with no mediastinal nodal involvement, a feature that might be helpful to differentiate LG from other disorders. (**c**) Interim Axial FDG/ PET/CT was obtained later and shows more numerus hypermetabolic renal masses (white arrows) and left anterior abdominal wall musculature masses (thin white arrow). (**d**) Microscopic picture of lymphomatoid granulomatosis showing angio-centric lesion with an infiltrate that is composed of a mixture of reactive appearing lymphocytes, plasma cells, and large transformed neoplastic cells. Note the inset shows the large, atypical cells are positive for in situ hybridization for EBV (dark blue stain at the periphery of the lesion). (Original magnification 400×, H&E stain; inset is EBER in situ hybridization stain).

**Figure 10 cancers-13-05853-f010:**
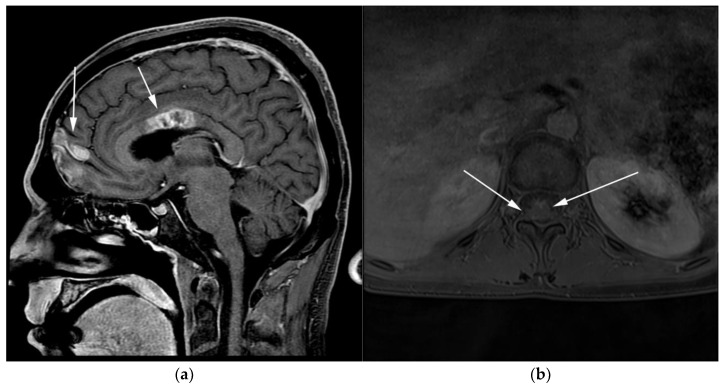
Imaging in the same patient 1 year later—patient developed new brain and spine lesions. (**a**) Sagittal T1 post-contrast MRI images show heterogenous mass within the body of the corpus callosum (posterior arrows) as well as within the frontal cortex (anterior arrows). (**b**) Axial T1 post-contrast MRI of the spine shows ill-defined enhancement within the spinal cord (arrows), consistent with leptomeningeal disease.

**Figure 11 cancers-13-05853-f011:**
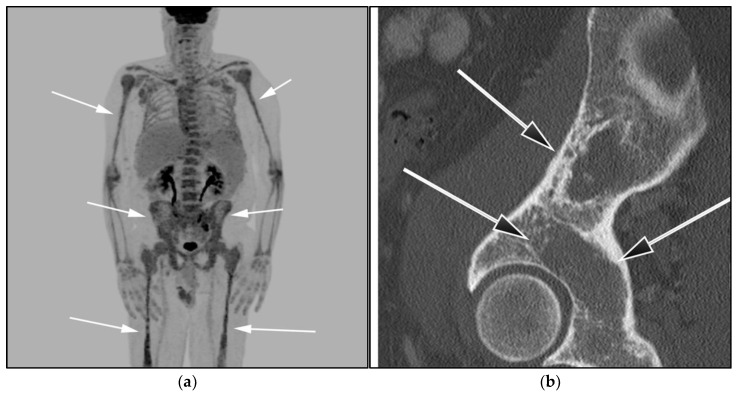
A 85-year-old-male with known diagnosis of lymphoplasmacytic lymphoma (LPL). (**a**) Maximal intensity projection (MIP) FDG PET/CT image shows extensive hypermetabolic uptake within the spine, pelvis, and proximal upper and distal appendicular skeleton (white arrows). This patient had not received recent marrow stimulating drugs, which can produce imaging findings consistent with marrow infiltration (**b**). A sagittal CT image shows diffuse osteopenia, marrow space widening, and no discrete focal lesions (black arrows). This is a very important to differentiate from myeloma, which often presents with frankly lytic lesions.

**Figure 12 cancers-13-05853-f012:**
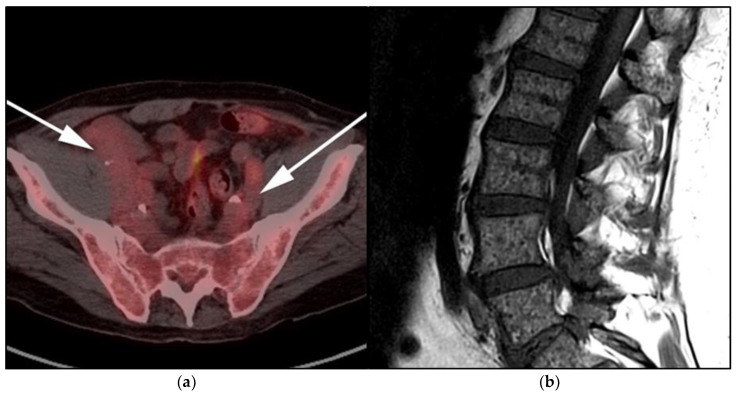
Lymphoplasmacytic lymphoma. (**a**) Axial FDG PET/CT in the same patient as Figure 8 shows hypermetabolic adenopathy within the bilateral iliac chain (white arrows). (**b**) Sagittal MRI T1-weighted image in the same patient shows a diffuse variegated appearance of the marrow consistent with marrow infiltration as that seen in multiple myeloma but without definite focal lytic lesions. Lymphoplasmacytic lymphoma combines features of both malignant lymphoma and multiple myeloma.

**Figure 13 cancers-13-05853-f013:**
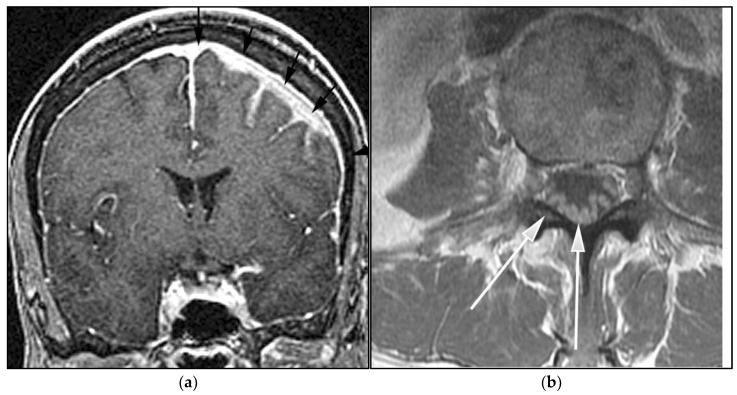
Patient with known diagnosis of lymphoplasmacytic lymphoma (LPL). (**a**) Coronal T1 post-contrast of the brain image shows diffuse leptomeningeal enhancement (black arrows). (**b**) Axial MRI T1 post-contrast image of the lumbosarcal spine in the same patient shows diffuse epidural soft tissue thickening and enhancement of the cauda equina nerve roots (white arrows). (**c**) Microscopic examination of the lymph node shows effaced architecture by infiltrative diffuse sheets of atypical lymphoid cells with plasmacytoid morphology. Tumor cells are positive for CD20 and are monotypic, expressing kappa or lambda light chains. (Original magnification 400×, H&E stain). CNS involvement in is an extremely rare variant known as Bing–Neel syndrome (BNS).

**Table 1 cancers-13-05853-t001:** Summary of changes contained in the updated 2016 WHO classification of B-cell LPD. New entities have been added, former entities have been recategorized, and some provisional entities have been “promoted”.

Disorders	Changes	New Diagnostic Implications
I-Premalignant conditions	Monoclonal B-cell lymphocytosis (MBL)—high cell count IgM monoclonal gammopathy of undetermined significance (MGUS) In situ follicular neoplasia (ISFN) In situ mantle cell neoplasia	Considered precursor for certain malignant lymphomas Now considered more closely related to B-cell lymphomas than multiple myeloma Periodic surveillance is recommended
II-Frank Lymphomas: A-Mature B-cell lymphoma (BCL)		
Lymphoplasmacytic lymphoma (LPL)	MYD88 L265P mutation present in majority of cases with changes in diagnostic criteria	Immunophenotype is mandatory to differentiate LPL, IgM myeloma, and IgM-secreting lymphomas such as marginal zone lymphoma (MZL) IgM MGUS is now believed to be more closely related to LPL and other IgM secreting B-cell lymphomas than to multiple myeloma
Waldenstrom macroglobinema (WM)		
IgM monoclonal gammopathy of undetermined significance (MGUS)		
Plasma cell myeloma	IgM MGUS is more closely related to LPL/WM than multiple myeloma	IgG MGUS is believed to be closely related to plasma cell myeloma-related disorders
Solitary plasmacytoma of the bone		
Extraosseous plasmacytoma		
Mantle cell lymphoma (MTL)	Currently believed to show greater heterogeneity in clinical behavior and phenotype than previously appreciated Two main types: − Leukemic − Classic MTL	Leukemic type: Indolent course frequently manifests as splenomegaly, bone marrow and peripheral blood involvement, and infrequent peripheral lymphadenopathy Classic type: Usually manifests as lymphadenopathy, bone marrow infiltration, and less commonly splenomegaly
Follicular lymphoma (FL) subtypes	Pediatric-type follicular lymphoma: now a definite entity, previously provisional entity Predominantly diffuse FL with 1p36 deletion	Pediatric FL is a localized form of FL affecting mainly children with predominant involvement of head and neck lymph nodes Favorable prognosis with little risk of progression following excision Aggressive, affecting mostly children and young adults Involves Waldeyer ring and/or cervical lymph nodes Accounts for some cases of diffuse FL, presents as localized mass, often inguinal nodes
Lymphomatoid granulomatosis (LG)	No changes	No changes
Diffuse large B-cell lymphoma (DLBCL)	Now mandatory to differentiate two DLBCL subtypes: including Germinal center B-cell (GCB) type and Activated B-cell type (ABC) Division has prognostic value, ABC compared with GCB lymphomas exhibit differential sensitivity to certain chemotherapy drugs, which may direct patient management	Prognosis is worse with ABC subtype
Primary diffuse large B-cell lymphoma of the CNS (PCNSL)	Prognosis has been updated Increased incidence has been confirmed	Baseline imaging is essential to differentiate PCNSL from secondary disease as part of disseminated DLBCL PCNSL is now believed to have poorer prognosis compared to systemic DLBCL 18F-FDG PET/CT is mandatory to differentiate primary vs. secondary CNS lymphoma
Primary mediastinal (thymic) B-cell lymphoma (PMLBCL) B-cell lymphoma unclassifiable, with features intermediate between DLBCL and classic Hodgkin lymphoma (CHL)	Cases that cannot be differentiated from CHL or more aggressive DLBCL are now categorized as B-cell lymphoma, unclassifiable, with features intermediate between DLBCL and CHL This category is used for lymphomas with overlapping clinical or morphological/immunophenotypic features between PMBCL, CHL, and DLBCL, formerly known as gray-zone lymphoma	Although the prognosis is better than DLBCL, it shows poorer prognosis compared to CHL
High-grade B-cell lymphoma with MYC and BCL2 and or BCL6 rearrangements	Double-hit lymphomas (DHLs)—majority of cases fall within the GCB subgroup, and tumors lacking chromosomal translocation and only expressing increased protein expression for BCL2 and MYC are called double expressor lymphomas This replaces the 2008 category of B-cell lymphoma, unclassifiable, with features intermediate between DLBCL and Burkitt lymphoma	Aggressive lymphomas with specific chromosomal translocations and/or protein expressions
High-grade B-cell lymphoma, NOS	Together with the new category for the “double-/triple-hit” lymphomas, replaces the 2008 category of B-cell lymphoma, unclassifiable, with features intermediate between DLBCL and Burkitt lymphoma Most cases lack MYC and BCL2 translocations	Aggressive DLBCL and BL not fitting any category
Burkitt-like lymphoma with 11q aberration	A new entity describing rare variant of high-grade B-cell lymphoma that closely resembles BL but lacks the MYC translocation and instead has frequent aberrations involving the 11q region	Mainly in children and are more often nodal than extranodal, in contrast to BL They are clinically aggressive but have a good response to therapy

Table references [5,6].

**Table 2 cancers-13-05853-t002:** Key pathologic, clinical, imaging features and differential diagnosis of rare B-cell lymphoma variants.

Entity	Pathologic Features	Clinical Features	Imaging Features	Differential Considerations
LC	NHL-BCL Most common subtype is DLBCL Prominent perivascular atypical large B cells in non-cohesive way Other subtypes: TCL Burkitt lymphoma	Rapid cognitive decline Psychiatric disturbances Gait disturbances Less commonly focal neurologic deficits	Bilateral diffuse abnormal T2-weighted hyperintensities along CST Poorly circumscribed non-mass-like lesions Lesions show avid metabolic activity on FDG-PET/CT	Infiltrative brain tumors, i.e., Gliomatosis cerebri Leukoencephalopathy Toxic/infectious demyelinating disorders
PMLBCL	Large BCL, thymic in origin Thick collagen bands around clusters of B cells producing compartmentalization Considered a subtype of DLBCL	Local mediastinal compressive symptoms Systemic manifestations mainly seen in relapsed disease	Heterogeneous hypermetabolic large ill-defined anterior mediastinal mass Supraclavicular nodes Systemic involvement seen in advanced disease	CHL Germ cell tumors Thymic tumors
LG	Mixture of reactive T cells, EBV-infected B cells, and plasma cells showing angio-destructive changes Histology ranges from a vasculitis-like pattern to sheets of large B cells	No history of allergen exposure No response to antimicrobial therapy Rapidly progressive with excellent response to corticosteroid therapy	Most common site is lung Other sites include renal, CNS Hypermetabolic bilateral diffuse reticulonodular opacities, mass-like No mediastinal nodes Rarely spleen, lymph nodes, and marrow	Pulmonary sarcoidosis LIP Atypical infections Hypersensitivity pneumonitis Wegner granulomatosis
LPL-WM	>10% mixed lymphoplasmacytic cells High IgM level deposition in tissues MYD88 ^L265P^ mutation in 95%, distinguishing from other overlapping B-cell lymphomas Immunophenotyping is mandatory to differentiate between IgM MM and IgM-secreting lymphomas High risk of transformation to DLBCL	Asymptomatic Hypercoagulability and easy bruising Raynaud phenomenon and/or ulcers in the lower extremities Extranodal infiltration	Bone marrow infiltration; however, unlike MM, no definite lytic lesions Hepatosplenomegaly Generalized lymphadenopathy CNS involvement BNS	IgM MGUS MM IgM-secreting B-cell lymphomas such as MALT, MZL

Abbreviations listed in Table 2: Lymphomatosis cerebri (LC), Non-Hodgkin’s lymphoma (NHL), B-Cell lymphoma (BCL), T-cell lymphoma (TCL), Diffuse large B-cell lymphoma (DBCL), Corticospinal tracts (CST), Primary mediastinal large B-cell lymphoma (PMLBCL), Classic Hodgkin’s lymphoma (CHL), Epstein–Barr virus (EBV), Lymphomatoid granulomatosis (LG), Lymphocytic interstitial pneumonitis (LIP), Wegner granulomatosis (WG), Lymphoplasmacytic lymphoma (LPL), Waldenstrom macroglobinema (WM), Multiple myeloma (MM), Bing–Neel syndrome (BNS), Monoclonal gammopathy of uncertain significance (MGUS), Mucosa-associated lymphoid tissue (MALT), Marginal zone lymphoma (MZL).

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
