# Peer review of "An Overview of Selected Rare B-Cell Lymphoproliferative Disorders: Imaging, Histopathologic, and Clinical Features"

_cancers, 2021, doi:10.3390/cancers13225853_

Round 1

Reviewer 1 Report

The authors significantly improved the manuscript, I feel now it presents and contains valuable information for physicians involved in the diagnostic process of these rare lymphoid tumors.

Author Response

We would like to thank the first reviewer for his/her comments. Additionally, the manuscript has been revised again for any spelling mistakes.

Reviewer 2 Report

The authors have satisfactorily addressed my comments. Here are some additional minor comments: 

1. Under follicular lymphoma subtypes (page 4), the correct name is "pediatric-type follicular lymphoma". 

2. Line 1532-1534: The distinction between WM and LPL is not based on the type of heavy chain secreted (IgM vs other). I pointed this out in my major comment no. 5. Please correct accordingly.   

Author Response

The authors have satisfactorily addressed my comments. Here are some additional minor comments: 

  1. Under follicular lymphoma subtypes (page 4), the correct name is "
    pediatric-type follicular lymphoma

Thanks for your comment, this has been changed.

  1. Line 1532-1534: The distinction between WM and LPL is not based on the type of heavy chain secreted (IgM vs other). I pointed this out in my major comment no. 5. Please correct accordingly.

Thanks for your comment, this has been removed. 

This manuscript is a resubmission of an earlier submission. The following is a list of the peer review reports and author responses from that submission.

Round 1

Reviewer 1 Report

In this review Salem et al. discuss the histopathologic, radiologic and clinical characteristics of rare lymphoid tumors (PCNSL, ALPS, PMBCL, lymphomatoid granulomatosis and lymphoplasmacytic lymphoma - WM).

I acknowledge that some types of these rare lymphomas have not yet been reviewed from the radiologist's perspective. I think therefore this article could be of interest for practicing radiologists. However, there are some errors and flaws in the manuscript that should be considered/discussed/corrected before publishing:

1) in line 64-65 it is stated: "PET/CT can further subclassify lymphomas into aggressive or indolent subtypes based on standardized uptake values (SUVs) and heterogeneity of disease involvement." Here I would like to note that some indolent lymphomas can present with a high SUV and with an indolent histomorphology (e.g. grade 1-2 FL). 

2) The sentence: "Aggressive lymphomas often display higher SUV (>10-13) on the baseline, or disease manifesting focally as increased metabolic activity in nodal or extranodal sites, or diffuse increased FDG uptake within extranodal sites" is unclear, I do not see the difference between the 3 types, hence all rely on high SUV/metabolic activity.

3) PET/CT cannot differentiate clonal tumors. For this, one should perform histological/molecular analysis. (lines 69-71)

4) This statement should be proved with at least one reference: "Lymphomas with suboptimal drop in SUVs, often relapses early even after achieving complete response."

5) It is stated several times that lymphoma transformation can be diagnosed with imaging. This is untrue, lymphoma transformation is histological diagnosis. Although, I acknowledge that higher SUV detected on the PET can raise suspicion, but this should be confirmed.

6) "After completion of therapy majority of aggressive lymphomas show residual masses" - in fact, aggressive lymphomas tend to respond to upfront immunochemo very well, around 2/3 of DLBCLs achieve at least initial CR..

7) There are several redundancies in lines between 127-162. It is stated more than one time that LC does not form definite tumor mass rather perivascular cuffs or that PCNSL usually presents with a focal solitary mass etc..

8) There are redundancies in the LPL - WM part, it is stated at least two times that WM secretes IgM meanwhile LPL secretes other forms if Ig. It is enough to mention this only once.

9) Sentence in line 451 should be merged with the previous one.

10) Legends of figure 8 and 10 are swapped.

11) Wegener granulomatosis is misspelled, and the term Granulomatosis with polyangiitis should be used.

12) "diagnosis of LPL is only reach after..." in lines 516-517 is incorrect grammatically.

Reviewer 2 Report

GENERAL COMMENTS

In this review the authors review the updates included in the most recent WHO classification of Lymphoid Neoplasms (2016) and discuss briefly 5 uncommon B-cell lymphoma variants (extremely rare or simply rare) or lymphoma-like disorders, namely Lymphomatosis cerebri (LC), autoimmune lymphoproliferative syndrome (ALPS), primary mediastinal large B-cell lymphoma (PMLBCL), lymphomatoid granulomatosis (LG), and lymphoplasmacytic lymphoma (LPL), Waldenstrom macroglobulinemia (WM) and Bing-Neel Syndrome, with respect to their clinical, histologic and imaging characteristics. Unfortunately this review article is not focused.

SPECIFIC COMMENTS

Major Comments

  1. The sections “Uncommon B-Cell-Lymphoma Variants and Lymphoma-like 2 disorders: Imaging and Histopathologic and Clinical Features” and “with Updates From the 4th Edition of The World Health Organization (WHO) Classification of Lymphoid Neoplasms”, as described in the title of the manuscript, are totally unrelated.
  2. It is not clear why the authors selected to discuss these particular 5 clinical entities, which are also totally unrelated, and not other rare entities. Some of them are not even included in the 2016 WHO classification.
  3. The 5 entities are not described systematically nor in a uniform way.
  4. Essential information is missing. For example, the diagnostic criteria for ALPS should be included.
  5. There are several inaccuracies included in the text: For example “In CHL, histologic features usually show abundant infiltration with lymphocytes with CD15 positivity” is inaccurate. The neoplastic cells are not conventional lymphocytes but the rare, large, Hodgkin and Reed-Sternberg cells. PMLBCL does not typically express CD15.
  6. Reference documentation is incomplete. For example, in page 13, lines 5-6, the authors cite a review instead of relatively large, recent studies on the prognostic factors in PMLBCL. Special mention is inappropriately made to paratracheal nodes in PMLBCL (it is not a specific feature) and the authors state that “Localized extension of PMLBCL to the supraclavicular nodes can occur, but further spread to other nodal stations is relatively uncommon.” Failing to recognize the predilection of the disease to infiltrate rare subdiaphragmatic extranodal sites (kidneys, adrenals etc) at diagnosis (uncommon) and more frequently at disease progression. In PMLBCL residual masses are a real problem much more frequently than rebound thymic hyperplasia and the authors do not adequately discuss this issue and the applicaton of PET/CT in this setting.
  7. The comments raised in points 4-6 (above) are more or less applicable to all the 5 entity description.

Minor Comments

The text includes several incomplete sentences.

Reviewer 3 Report

In this narrative review, the authors (Salem et al.) highlight changes in the 2016 WHO classification of lymphoid neoplasms and summarize the salient clinical, histological and radiologic features of some rare lymphoid disorders (primary CNS lymphoma, autoimmune lymphoproliferative disorder, lymphomatoid granulomatosis, primary mediastinal large B cell lymphoma and lymphoplasmacytic lymphoma). The review is a useful summary for practicing radiologists.

I have the following major and minor comments for the authors to consider –

Major:

  1. It is unclear to me why the authors decided to choose the above 5 disorders to focus on further. It would be helpful to the readers if the authors provide some rationale for why they chose to focus on these 5 disorders. Perhaps these disorders are particularly challenging for the radiologist or there have been substantial recent changes in the clinical diagnostic criteria. It would be highly desirable to connect these disorders with some common theme.
  2. On multiple occasions, the authors state that “MGUS is now believed to be more closely related to LPL and other IgM secreting B cell lymphomas than to multiple myeloma”. This is certainly true for IgM MGUS which is biologically similar to LPL/WM than multiple myeloma. However, this is not true for non-IgM MGUS which is indeed biologically similar to multiple myeloma rather than B cell lymphomas. Therefore, I would strongly suggest qualifying MGUS as IgM or non-IgM in the text.
  3. The description of CHL and its distinction from PMBLCL (lines 290-295) is inaccurate. Please revise accordingly. For example, CHL is described as showing “abundant infiltration with lymphocytes with CD15 positivity”. A more accurate description would be – “rare Hodgkin-Reed Sternberg (HRS) cells are present in a polymorphous background composed of small lymphocytes, macrophages, eosinophils and plasma cells. The HRS cells typically show expression of CD30 and CD15 and lack expression of CD20 and CD45”
  4. The definition of LPL and its distinction from WM is inaccurately described in lines 425-427. Lymphoplasmacytic lymphomas most commonly secrete an IgM paraprotein and this does not preclude a diagnosis of LPL. So the diagnosis of LPL is not reserved only for tumors that secrete non-IgM paraproteins.

Minor:

  1. Table 1, under PMLBCL (end of page 4) the authors state “Cases that cannot be differentiated from CHL or more aggressive DLBCL by genetic profiling is now categorized as B cell lymphoma, unclassifiable, with fea-tures intermediate between DLBCL and CHL”. I would suggest removing “genetic profiling” as it is unclear if genetic testing is sufficient to make this distinction. Per WHO criteria, this distinction is made on clinical, morphologic and immunophenotypic grounds. Also, the description appears to be for the cateogory B cell lymphoma unclassifiable with features intermediate between DLBCL and CHL but PMLBCL is also listed; is the latter an error?
  2. Table 1, under double hit lymphoma – “Double-hit lymphomas (DHLs) fall within the GCB subgroup”. While the majority of cases fall within the GCB subgroup, many cases are not. So I would modify the statement accordingly.
  3. Table 1, under high grade B cell lymphoma, NOS “lacks MYC and BCL2 translocations”. This is incorrect. The presence of a MYC rearrangement does not preclude this diagnosis. A MYC rearrangement can indeed be identified in cases classified as such. There is also a typo in “Aggressive DLBCL and BL nit fitting any category”
  4. Table 2, under PMLBCL – “previously considered a subtype”. In the current WHO classification system, PMLBCLs are considered a special subtype of diffuse large B cell lymphoma.
  5. Table 2, under LG, - “Ranges for vasculitis to DLBCL” I would suggest modifying this sentence to “histology ranges from a vasculitis-like pattern to sheets of large B cells”
  6. Table 2, under LPL-WM – “High risk of transformation to MM or DLBCL”. While cases of LPL can transform to DLBCL, transformation to multiple myeloma is not a recognized phenomenon.
  7. Table 2, under LPL-WM – fragmented sentence in “Bone marrow infiltration in”
  8. Line 268: “There is marked expansion of sinus histiocytes lacking Langerhans-type features”. It is sufficient to describe this pattern as “sinus histiocytosis”.